# Broad-Coverage Semantic Parsing: A Transition-Based Approach

## Abstract

The representation of many common semantic phenomena requires structural properties beyond those commonly used for syntactic parsing. We discuss a set of structural properties required for broad-coverage semantic representation, and note that existing parsers support some of these properties, but not all. We propose two transition-based techniques for parsing such semantic structures: (1) applying conversion procedures to map them into related formalisms, and using existing state-of-the-art parsers on the converted representations; and (2) constructing a parser that directly supports the full set of properties. We experiment with UCCA-annotated corpora, the only ones with all these structural semantic properties. Results demonstrate the effectiveness of transition-based methods for the task.

## 1 Introduction

In order for a grounded semantic representation to cover the full range of semantic structures exhibited by natural language, there are three structural properties that should be supported. The first is **multiple parents**, representing arguments and relations (semantic units) that are shared between predicates. For instance, in the sentence "After graduation, John moved to Paris", "John" is an argument of both "graduation" and "moved", yielding a DAG structure (Figure 1a), rather than a tree.

The second is **non-terminal nodes** for representing units comprising more than one word. While bi-lexical dependencies partially circumvent this requirement, by representing complex units in terms of their headwords, they fall short when representing units that have no clear head.

Frequent examples of such constructions include coordination structures (e.g., "*John and Mary* went home"; Figure 1b), some multi-word expressions (e.g., "The Haves and the *Have Nots*"), and prepositional phrases. In these cases, dependency schemes often apply some convention selecting one of the sub-units as the head, but as different head selections are needed for different purposes, standardization problems arise (Ivanova et al., 2012). For example, selecting the preposition to head prepositional phrases yields better parsing results (Schwartz et al., 2012), while the head noun is more useful for information extraction.

Third, semantic units may be **discontinuous** in the text. For instance, in "John *gave* everything *up*" (Figure 1c), the phrasal verb "gave ... up" forms a single semantic unit. Discontinuities are also pervasive with other multi-word expressions (Schneider et al., 2014). We call formal representations supporting all three properties *Broad-coverage Semantic Structures* (BSS).

However, to our knowledge, no existing parser for a grounded semantic annotation scheme supports the combination of these criteria. The only such scheme supporting them is UCCA (Abend and Rappoport, 2013), which has no parser. Several other models either support some of these properties (Oepen et al., 2015), or avoid grounding semantic units altogether (notably, AMR (Banarescu et al., 2013); see Section 2).

In this work we are first in proposing techniques for BSS parsing. We adopt a transition-based approach, which has recently produced some of the best results in syntactic dependency parsing (Dyer et al., 2015; Ballesteros et al., 2015), and has also demonstrated strong performance in a variety of other semantic and syntactic settings (Maier, 2015; Wang et al., 2015, among others). Transition-based methods are a natural starting point for UCCA parsing, as the set of distinctions

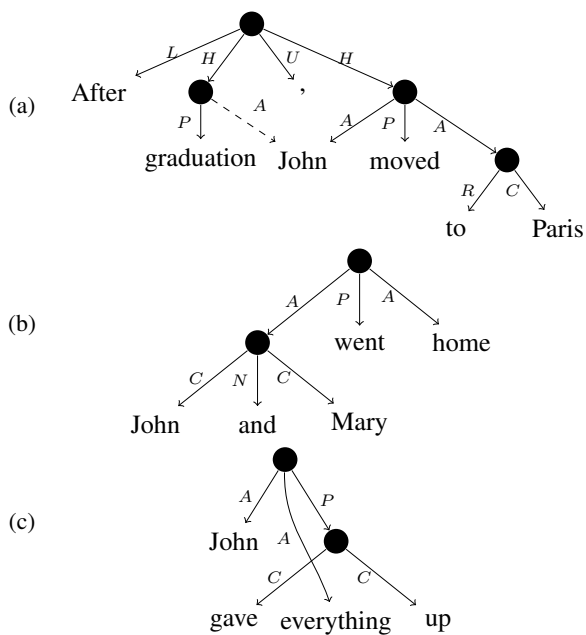

Figure 1: Semantic representation of the three structural properties required for BSS, according to the UCCA scheme. (a) includes a remote edge (dashed), resulting in "John" having two parents. (b) includes a coordination construction ("John and Mary"). (c) includes a discontinuous unit ("gave ... up"). Legend: $P$ – a Scene's main relation, $A$ – participant, $L$ – inter-scene linker, $H$ – linked Scene, $C$ – center, $R$ – relator, $N$ – connector, $U$ – punctuation, $F$ – function unit. Pre-terminal nodes are omitted for brevity.

it represents, centered around predicate-argument structures and their inter-relations, is similar to distinctions conveyed by dependency schemes.

We pursue two complementary parsing strategies. First, we assess the ability of existing technology to tackle the task, by developing conversion protocols between UCCA structures and two related formalisms: dependency trees and discontinuous constituency trees. As these formalisms are more restrictive than BSS, the conversion is necessarily lossy. Nonetheless, we find that it is effective in practice (Section 3). Second, we present a novel transition-based broad-coverage parser, Broad-coverage Semantic Parser (BSP; Section 4) supporting multiple parents, non-terminal nodes and discontinuous units, based on extending existing transition-based parsers with new transitions and features.

We experiment with the English UCCA-annotated corpora (Abend and Rappoport, 2013) as a test case, in both in-domain and out-of-domain scenarios, reaching nearly 70% labeled F-score for the highest scoring parser. The results suggest concrete paths for further improvement. All converters and parsers will be made publicly available upon publication.

## 2 Background

**Broad-coverage Semantic Representation.** While earlier work on grounded semantic parsing has mostly concentrated on shallow semantic analysis, focusing on semantic role labeling of verbal argument structures, the focus has recently shifted to parsing of more elaborate representations that account for a wider range of phenomena. Most closely related to this work is Broad-coverage Semantic Dependency Parsing (SDP), addressed in two SemEval tasks (Oepen et al., 2014; Oepen et al., 2015), experimenting with the Prague tectogrammatical layer (Sgall et al., 1986; Böhmová et al., 2003), and with dependencies derived from the Enju parser,[1] and Lingo ERG (Flickinger, 2002). Like BSS parsing, SDP addresses a wide range of semantic phenomena, and supports discontinuous units and multiple parents. However, SDP uses bi-lexical dependencies, disallowing non-terminal nodes, and thus faces difficulties in supporting structures that have no clear head, such as coordination (Ivanova et al., 2012).

Another line of work addresses parsing into non-grounded[2] semantic representation, notably Abstract Meaning Representation (AMR) (Flanigan et al., 2014; Vanderwende et al., 2015; Pust et al., 2015; Artzi et al., 2015). While sharing much of this work's motivation, not grounding the representation in the text complicates the parsing task, as it requires that the alignment between words and logical symbols be automatically (and imprecisely) detected.[3] Furthermore, grounding allows breaking down sentences into semantically meaningful sub-spans, which is useful for many applications (see discussion in Fernández-González and Martins (2015)). Wang et al. (2015) applied a transition-based approach to AMR parsing. Their method involved first syntactically parsing the input, and then converting the result into AMR. Other approaches for semantic representation, such as MRS (Copestake et al., 2005) and DRT (Kamp et al., 2011), involve considerably different representation and parsing approaches, and so fall beyond the scope of our discussion.

---

[1]See http://kmcs.nii.ac.jp/enju

[2]By *grounded* we mean the text tokens are directly annotated as part of the representation, as opposed to abstract formalisms approximating logical form, for example.

[3]Considerable technical effort has been invested in the AMR alignment task under various approaches (Flanigan et al., 2014; Pourdamghani et al., 2014; Pust et al., 2015).

**The UCCA Annotation Scheme.** Universal Cognitive Conceptual Annotation (UCCA) is a cross-linguistically applicable semantic representation scheme, that builds on the established "Basic Linguistic Theory" framework for typological description (Dixon, 2010a; Dixon, 2010b; Dixon, 2012), and on the Cognitive Linguistics literature. UCCA is a multi-layered representation, where each layer corresponds to a "module" of semantic distinctions (e.g., predicate-argument structures, adverbials, coreference etc.).

Formally, a UCCA structure over a sentence is a DAG, whose leaves correspond to the sentence's words. The nodes of the graph, or its "units", are either terminals or several sub-units (not necessarily contiguous) jointly viewed as a single entity according to some semantic or cognitive consideration. Edges bear a category, indicating the role of the sub-unit in the relation that the parent represents. UCCA structures support all three criteria of BSS: multiple parents, non-terminal nodes, and discontinuous units.

UCCA's foundational layer, which we use here, covers the predicate-argument structures evoked by predicates of all grammatical categories (verbal, nominal, adjectival and others), the interrelations between them, as well as other major linguistic phenomena such as coordination and multi-word expressions. This set of categories has demonstrated applicability to multiple languages, including English, French, German and Czech, support for rapid annotation, and semantic stability in translation: UCCA annotations of translated text usually contain the same set of relationships (Sulem et al., 2015). This finding supports the claim that UCCA represents an abstract level of semantics, shared by different languages.

The layer's basic notion is the *Scene*, which describes a movement, action or state. Each Scene contains one Main Relation, as well as one or more Participants. For example, the sentence "After graduation, John moved to Paris" contains two Scenes, whose main relations are "graduation" and "moved". "John" is a Participant in both Scenes, while "Paris" only in the latter. UCCA marks one of the incoming edges for each non-root as "primary" and the others as "remote" edges. The two Scenes in this sentence are both arguments of the *Linker* "After", which in this case expresses a temporal relation between them. Figure 1 presents the UCCA annotation of this and other examples.

Further categories account for relations between Scenes and the internal structures of complex arguments (e.g., coordination) and relations (e.g., complex adverbials, such a "very clearly"). UCCA graphs may contain implicit units that have no correspondent in the text, but the parsing of these units is deferred to future work, as it is likely to require different methods than those explored here (Roth and Frank, 2015).

## 3 Conversion-Based Parsing

We begin by assessing the ability of existing technology to address the task, by taking a conversion-based approach. Training proceeds by converting BSS into a different representation, and training an existing parser on the converted structures. We evaluate the trained parsers by applying them to the test set, and converting the results back to BSS, where they are compared with the gold standard. The error resulting from this back and forth conversion is discussed in Section 6.

**Notation.** Let $L$ be the set of possible edge labels. A BSS is a directed acyclic graph $G = (V, E, \ell)$ over a sequence of tokens $w_1, \ldots, w_n$, where $\ell : E \to L$ is a function of *edge labels*. For each token $w_i$ $(i = 1, \ldots, n)$, there exists a leaf (or a terminal) $t_i \in V$.

**Conversion to Constituency Trees.** We convert BSS to constituency trees by removing a subset of the edges.[4] Specifically, when converting UCCA structures, we simply remove all remote edges, leaving only primary edges, which form a tree structure (see Figure 1a). The inverse conversion from trees to BSS is simply the identity function, as every constituency tree is a BSS.

**Conversion to Dependency Trees.** In the conversion to dependency trees, we first convert BSS to constituency trees using the above procedure, and then convert the result to dependency trees. Assume $T_c = (V_c, E_c, \ell_c)$ is a constituency tree with labels $\ell_c : E_c \to L$, where $L$ is the set of possible labels. The conversion from $T_c$ to a dependency tree involves the removal of all non-terminals from $T_c$ and the addition of edges between terminals. The nodes of the converted dependency tree are simply the terminals of $T_c$.

We define a linear order over possible edge labels $L$. For each node $u \in V$, denote with

---

[4] For trees, labeling nodes is equivalent to labeling edges. Thus, we do not distinguish between the two options. Note also that as the original structures may contain discontinuities, so may the resulting trees.

**Data:** constituency tree $T_c = (V_c, E_c, \ell_c)$
**Result:** dependency tree $T_d = (V_d, E_d, \ell_d)$
**foreach** $u \in V_c$ **do**
 | $h(u) \leftarrow \text{argmin}_v \text{Priority}(\ell_c(u, v))$;
**end**
$V_d \leftarrow \text{Terminals}(T_c)$, $E_d \leftarrow \emptyset$;
**foreach** $t \in V_d$ **do**
 | $u \leftarrow t$;
 | **while** $u = h(\text{Parent}_c(u))$ **do**
 | $h^*(u) \leftarrow t$;
 | $u \leftarrow \text{Parent}_c(u)$;
 | **end**
 | $n(t) \leftarrow h(u)$;
**end**
**foreach** $t \in V_d$ **do**
 | $u \leftarrow \text{Parent}_c(n(t))$;
 | $t' \leftarrow h^*(u)$;
 | $E_d \leftarrow E_d \cup \{(t', t)\}$;
 | $\ell_d(t', t) \leftarrow \ell_c(u, n(t))\}$;
**end**

Algorithm 1: Constituency to dependency conversion procedure.

**Data:** dependency tree $T_d = (V_d, E_d, \ell_d)$
**Result:** constituency tree $T_c = (V_c, E_c, \ell_c)$
$r \leftarrow \text{Node}()$;
$V_c \leftarrow \{r\}$, $E_c \leftarrow \emptyset$;
**foreach** $t \in \text{TopologicalSort}(V_d)$ **do**
 | $u \leftarrow \text{Node}()$;
 | $V_c \leftarrow V_c \cup \{u, t\}$;
 | $E_c \leftarrow E_c \cup \{(u, t)\}$;
 | $\ell_c(u, t) \leftarrow Terminal$;
 | $t' \leftarrow \text{Parent}_d(t)$;
 | **if** $t' = \text{ROOT}$ **then**
 | $E_c \leftarrow E_c \cup \{(r, u)\}$;
 | $\ell_c(r, u) \leftarrow \text{Label}(r)$;
 | **else**
 | **if** $\exists v \in V_d : (t, v) \in E_d$ **then**
 | $u' \leftarrow \text{Node}()$;
 | $E_c \leftarrow E_c \cup \{(u', u)\}$;
 | $\ell_c(u', u) \leftarrow \text{Label}(u')$;
 | **else**
 | $u' \leftarrow u$;
 | **end**
 | $p \leftarrow \text{Parent}_c(t')$;
 | $E_c \leftarrow E_c \cup \{(p, u')\}$;
 | $\ell_c(p, u') \leftarrow \ell_d(t', t)$;
 | **end**
**end**

Algorithm 2: Dependency to constituency conversion procedure.

$h(u)$ its child with the highest edge label. Denote with $h^*(u)$ the terminal reached by recursively applying $h(\cdot)$ over $u$. For each terminal $t$, we define $n(t)$ as the highest non-terminal such that $t = h^*(n(t))$, i.e., $n(t)$ is the only node such that $t = h^*(n(t))$ and $t \neq h^*(\text{Parent}_c(n(t)))$. The head of $t$ according to the dependency graph is the terminal $h^*(\text{Parent}_c(n(t)))$. The complete conversion procedure from constituency to dependency is given in Algorithm 1.

We note that this conversion procedure is simpler than the head percolation procedure used for converting syntactic constituency trees to dependency trees. This is because $h(u)$ of a node $u$ (similar to $u$'s head-containing child), depends only on the label of the edge $(h(u), u)$, and not on the subtree spanned by $u$, because edge labels in UCCA directly express the role of the child in the parent unit, and are thus sufficient for determining which of $u$'s children contains the head node.

The inverse conversion introduces non-terminal nodes back into the tree. As the distinction between low- and high-attaching nodes is lost in the constituency to dependency conversion, we heuristically assume that attachments are always high-attaching. Assume $T_d = (V_d, E_d, \ell_d)$ is a dependency tree. We begin by creating a root node $r$. Then, iterating over $V_d$ in topological order, we add its members as terminals to the constituency tree and create a pre-terminal parent for each, with an edge labeled as *Terminal* between them. The parents of the pre-terminals are determined by the terminal's parent in the dependency tree: if a dependency node $t$ is a child of the root in $T_d$, then $t$'s pre-terminal will also be a child of the root node. Otherwise, $t$'s pre-terminal is the child of the pre-terminal associated with $t$'s head in $T_d$. We add an intermediate node in between if $t$ has any dependents in $T_d$, to allow adding their pre-terminals as children. Edge labels for the intermediate edges are determined by a rule-based function, denoted by $\text{Label}(u)$.[5] In practice, it mostly selects the UCCA label *Center*. This conversion procedure is given in Algorithm 2.

# 4 Broad-Coverage Semantic Parsing

We now turn to presenting BSP, a transition-based parser that supports the three criteria of broad-coverage semantic structures.

Transition-based parsing (Nivre, 2003) creates the parse as it scans the text from left to right. The parse is created incrementally by applying a *transition* at each step to the parser state, defined using three data structures: a buffer $B$ of tokens and nodes to be processed, a stack $S$ of nodes currently being processed, and a graph $G = (V, E, \ell)$ of constructed nodes and labeled edges. Some of the states are marked as *terminal*, meaning that $G$ is the final output. A classifier is used at each step

---

[5] See Supplementary Material for the definition of Label.

to select the next transition based on features that encode the parser's current state. During training, an oracle creates training instances for the classifier, based on the gold-standard annotation.

Despite being based on local decisions, transition-based methods have yielded excellent results in a variety of parsing tasks. Within syntactic dependency parsing, transition-based methods have been successfully applied to corpora in many languages and domains, yielding some of the best reported results (Dyer et al., 2015; Ballesteros et al., 2015). The approach has also yielded results comparable with the state-of-the-art in constituency parsing (Sagae and Lavie, 2005; Zhang and Clark, 2009; Zhu et al., 2013), discontinuous constituency parsing (Maier, 2015), as well as dependency DAG structures (Sagae and Tsujii, 2008; Tokgöz and Eryiğit, 2015), CCG structures (Ambati et al., 2015) and AMR parsing (Wang et al., 2015).

BSP mostly builds on recent advances in discontinuous constituency and dependency DAG parsing techniques, and further introduces novel UCCA-oriented features for parsing BSS.

**Transition Set.** Given a sequence of tokens $w_1, \ldots, w_n$, we predict a BSS $G$ whose leaves correspond to the tokens. Parsing starts with a single node on the stack (the root node), and the input tokens $w_1, \ldots, w_n$ in the buffer. The set of transitions is given in Figure 2. In addition to the standard SHIFT and REDUCE operations, we follow previous work in transition-based constituency parsing (Sagae and Lavie, 2005), and include the NODE transition for creating new non-terminal nodes. Concretely, NODE$_X$ creates a new node on the buffer as a parent of the first element on the stack, with an $X$-labeled edge.

LEFT-EDGE$_X$ and RIGHT-EDGE$_X$ create a new primary $X$-labeled edge between the first two elements on the stack, where the parent is the left or the right node, respectively. As a UCCA node may only have one incoming primary edge, EDGE transitions are disallowed where the child node already has an incoming primary edge. LEFT-REMOTE$_X$ and RIGHT-REMOTE$_X$ do not have this restriction, and the created edge is marked as *remote*. We distinguish between these two pairs of transitions, for the parser to be able to determine whether an edge is a primary or a remote one. In order to support the prediction of multiple parents, node and edge transitions do not automatically ap-

ply REDUCE. This is in line with other work on transition-based DAG dependency parsing (Sagae and Tsujii, 2008; Tokgöz and Eryiğit, 2015). Once all edges for a particular node have been created, it is removed from the stack by applying REDUCE.

SWAP allows handling discontinuous nodes, by popping the second node on the stack and adding it to the top of the buffer, as with the similarly named transition in previous work (Nivre, 2009; Maier, 2015). Finally, FINISH pops the root node and marks the state as terminal.

**Features.** Figure 3 presents the feature templates used by the parser. For some of the features, we used the notion of *head word*, defined by the $h^*(\cdot)$ function (Section 3). While head words are not explicitly represented in the UCCA scheme, these features proved useful as means of encoding word-to-word relations.

In addition to the binary features defined by the feature templates, we employ a real-valued feature, **ratio**, corresponding to the ratio between the number of terminals to number of nodes in the graph $G$. This novel feature serves as a regularizer for the creation of new nodes, and should be beneficial for other transition-based constituency parsers too. Features are generally adapted from the related parsers of (Zhang and Clark, 2009; Zhu et al., 2013; Tokgöz and Eryiğit, 2015; Maier, 2015), with a small additional set of features encoding relevant information for the novel LEFT-REMOTE$_X$ and RIGHT-REMOTE$_X$ transitions.

**Training.** Following Maier (2015), we use a linear classifier, using the averaged structured perceptron algorithm for training it (Collins and Roark, 2004) with the MINUPDATE (Goldberg and Elhadad, 2011) procedure: a minimum number of updates to a feature has to occur in training for it to be included in the model. Inference is performed greedily (i.e., without beam search).

For training the local classifier, we use a dynamic oracle (Goldberg and Nivre, 2012), i.e., an oracle that outputs a set of optimal transitions: when applied to the current parser state, the gold standard graph is reachable from the resulting state. For example, the oracle would predict a NODE transition if the stack has on its top a parent in the gold graph that has not been created, but would predict a RIGHT-EDGE transition if the second stack element is a parent of the first element according to the gold graph and the edge between them has not been created. The transition

| | Initial State | | | | | Final State | | | |
|---|---|---|---|---|---|---|---|---|---|
| **Stack** | **Buffer** | **Nodes** | **Edges** | **Terminal?** | **Stack** | **Buffer** | **Nodes** | **Edges** | **Terminal?** |
| [root] | $w_{1:n}$ | $\{\text{root}\} \cup$ $w_{1:n}$ | $\emptyset$ | $-$ | $\emptyset$ | $\emptyset$ | $V$ | $E$ | $+$ |

| | Before Transition | | | Transition | | After Transition | | | | Condition |
|---|---|---|---|---|---|---|---|---|---|---|
| **Stack** | **Buffer** | **Nodes** | **Edges** | | **Stack** | **Buffer** | **Nodes** | **Edges** | **Terminal?** | |
| $S$ | $x \mid B$ | $V$ | $E$ | SHIFT | $S \mid x$ | $B$ | $V$ | $E$ | $-$ | |
| $S \mid x$ | $B$ | $V$ | $E$ | REDUCE | $S$ | $B$ | $V$ | $E$ | $-$ | |
| $S \mid x$ | $B$ | $V$ | $E$ | NODE$_X$ | $S \mid x$ | $y \mid B$ | $V \cup \{y\}$ | $E \cup \{(y,x)_X\}$ | $-$ | $x \neq \text{root}$ |
| $S \mid y,x$ | $B$ | $V$ | $E$ | LEFT-EDGE$_X$ | $S \mid y,x$ | $B$ | $V$ | $E \cup \{(x,y)_X\}$ | $-$ | $\begin{cases} x \notin w_{1:n}, \\ y \neq \text{root}, \\ y \not\rightsquigarrow_G x \end{cases}$ |
| $S \mid x,y$ | $B$ | $V$ | $E$ | RIGHT-EDGE$_X$ | $S \mid x,y$ | $B$ | $V$ | $E \cup \{(x,y)_X\}$ | $-$ | |
| $S \mid y,x$ | $B$ | $V$ | $E$ | LEFT-REMOTE$_X$ | $S \mid y,x$ | $B$ | $V$ | $E \cup \{(x,y)_X^*\}$ | $-$ | |
| $S \mid x,y$ | $B$ | $V$ | $E$ | RIGHT-REMOTE$_X$ | $S \mid x,y$ | $B$ | $V$ | $E \cup \{(x,y)_X^*\}$ | $-$ | |
| $S \mid x,y$ | $B$ | $V$ | $E$ | SWAP | $S \mid y$ | $x \mid B$ | $V$ | $E$ | $-$ | $i(x) < i(y)$ |
| [root] | $\emptyset$ | $V$ | $E$ | FINISH | $\emptyset$ | $\emptyset$ | $V$ | $E$ | $+$ | |

Figure 2: The transition set of BSP. We write the stack with its top to the right and the buffer with its head to the left. $(\cdot,\cdot)_X$ denotes a primary $X$-labeled edge, and $(\cdot,\cdot)_X^*$ a remote $X$-labeled edge. $i(x)$ is a running index for the created nodes. EDGE transitions have an additional condition: the prospective child may not already have a primary parent.

predicted by the classifier is deemed correct and is applied to the parser state to reach the subsequent state, if the transition is included in the set of optimal transitions. Otherwise, a random optimal transition is applied and the classifier's weights are updated according to the perceptron update rule.

## 5 Experimental Setup

**Data.** We conduct our main experiments on the UCCA Wikipedia corpus (henceforth, *Wiki*), and use the English part of the UCCA *Twenty Thousand Leagues Under the Sea* English-French parallel corpus (henceforth, *20K Leagues*) as out-of-domain data.[6] Table 1 presents some statistics for the two corpora, demonstrating that while the *Wiki* corpus is over ten times larger, the overall statistics are similar. We use passages of indices up to 655 of the *Wiki* corpus as our training set, passages 656–700 as development set, and passages 701–695 as in-domain test set. While UCCA edges can cross sentence boundaries, we adhere to the common practice in semantic parsing and train our parsers on individual sentences, discarding inter-relations between them (0.18% of the edges). We also discard linkage nodes and edges, as they often express inter-sentence relations and are thus mostly redundant when applied at the sentence level, as well as implicit nodes (Section 2). In the out-of-domain experiments, we apply the same parser (trained on the *Wiki* corpus) to the *20K Leagues* corpus without re-tuning the parameters.

**Evaluation.** Since there are no standard evaluation measures for BSS, we define two simple

---

[6]Both are available at http://www.cs.huji.ac.il/~oabend/ucca.html

| | Wiki | | | 20K |
|---|---|---|---|---|
| | Train | Dev | Test | Leagues |
| # passages | 281 | 35 | 43 | 154 |
| # sentences | 4021 | 537 | 608 | 522 |
| # nodes | 277,587 | 40,700 | 45,047 | 29,965 |
| % terminal | 42.41 | 42.8 | 42.66 | 41.23 |
| % non-term. | 57.59 | 57.20 | 57.34 | 58.77 |
| % implicit | 0.29 | 0.35 | 0.27 | 0.8 |
| % linkage | 0.92 | 0.96 | 0.9 | 1.25 |
| % discont. | 0.52 | 0.55 | 0.47 | 0.79 |
| % >1 parent | 2.29 | 1.89 | 2.21 | 1.98 |
| # edges | 272,018 | 39,660 | 44,139 | 28,723 |
| % primary | 95.37 | 95.70 | 95.90 | 94.48 |
| % remote | 1.69 | 1.24 | 1.32 | 2.19 |
| % linkage | 2.94 | 3.06 | 2.78 | 3.33 |
| Average per non-linkage non-terminal node | | | | |
| # children | 1.67 | 1.67 | 1.67 | 1.61 |

Table 1: Statistics of the *Wiki* and *20K Leagues* UCCA corpora. All counts exclude the root node.

measures for comparing such structures. Assume $G_p = (V_p, E_p, \ell_p)$ and $G_g = (V_g, E_g, \ell_g)$ are the predicted and gold-standard DAGs over the same sequence of terminals $W = \{w_1, \ldots, w_n\}$, respectively. For an edge $e = (u, v)$ in either graph, where $u$ is the parent and $v$ is the child, define its yield $y(e) \subseteq W$ as the set of terminals in $W$ that are descendants of $v$. We define the set of *mutual edges* between $G_p$ and $G_g$:

$$M(G_p, G_g) =$$
$$\{(e_1, e_2) \in E_p \times E_g \mid y(e_1) = y(e_2) \wedge \ell_p(e_1) = \ell_g(e_2)\}$$

Labeled precision and recall are defined by dividing $|M(G_p, G_g)|$ by $|E_p|$ and $|E_g|$, respectively. We report two variants of this measure, one where we consider only non-remote edges, and another where we consider remote edges. We note that the measure collapses to the standard PARSEVAL constituency evaluation measure if $G_p$ are

Features from (Zhang and Clark, 2009):

**unigrams**

$s_0te, s_0we, s_1te, s_1we, s_2te, s_2we, s_3te, s_3we,$

$b_0wt, b_1wt, b_2wt, b_3wt,$

$s_0lwe, s_0rwe, s_0uwe, s_1lwe, s_1rwe, s_1uwe$

**bigrams**

$s_0ws_1w, s_0ws_1e, s_0es_1w, s_0es_1e, s_0wb_0w, s_0wb_0t,$

$s_0eb_0w, s_0eb_0t, s_1wb_0w, s_1wb_0t, s_1eb_0w, s_1eb_0t,$

$b_0wb_1w, b_0wb_1t, b_0tb_1w, b_0tb_1t$

**trigrams**

$s_0es_1es_2w, s_0es_1es_2e, s_0es_1eb_0w, s_0es_1eb_0t,$

$s_0es_1wb_0w, s_0es_1wb_0t, s_0ws_1es_2e, s_0ws_1eb_0t$

**separator**

$s_0wp, s_0wep, s_0wq, s_0wcq, s_0es_1ep, s_0es_1eq,$

$s_1wp, s_1wep, s_1wq, s_1weq$

**extended** (Zhu et al., 2013)

$s_0llwe, s_0lrwe, s_0luwe, s_0rlwe, s_0rrwe,$

$s_0ruwe, s_0ulwe, s_0urwe, s_0uuwe, s_1llwe,$

$s_1lrwe, s_1luwe, s_1rlwe, s_1rrwe, s_1ruwe$

**disco** (Maier, 2015)

$s_0xwe, s_1xwe, s_2xwe, s_3xwe,$

$s_0xte, s_1xte, s_2xte, s_3xte,$

$s_0xy, s_1xy, s_2xy, s_3xy$

$s_0xs_1e, s_0xs_1w, s_0xs_1x, s_0ws_1x, s_0es_1x,$

$s_0xs_2e, s_0xs_2w, s_0xs_2x, s_0ws_2x, s_0es_2x,$

$s_0ys_1y, s_0ys_2y, s_0xb_0t, s_0xb_0w$

Features from (Tokgöz and Eryiğit, 2015):

**counts**

$s_0P, s_0C, s_0wP, s_0wC, b_0P, b_0C, b_0wP, b_0wC$

**edges**

$s_0s_1, s_1s_0, s_0b_0, b_0s_0, s_0b_0e, b_0s_0e$

**history**

$a_0we, a_1we$

**remote** (Novel, UCCA-specific features)

$s_0R, s_0wR, b_0R, b_0wR$

Figure 3: Feature templates for BSP. Notation: $s_i$, $b_i$ are the $i$th stack and buffer items, respectively. $w$ and $t$ are the word form and part-of-speech tag of the terminal returned by the $h^*(\cdot)$ function (Section 3). $e$ is the edge label to the node returned by the $h(\cdot)$ function. $l$, $r$ ($ll$, $rr$) are the left-most and rightmost (grand)children, respectively. $u$ ($uu$) is the unary (grand)child, when only one exists. $p$ is a unique separator punctuation and $q$ is the separator count between $s_0$ and $s_1$. $x$ is the gap type ("none", "pass" or "gap") at the sub-graph under the current node, and $y$ is the sum of gap lengths (Maier and Lichte, 2011). $P$ and $C$ are the number of parents and children, respectively, and $R$ is the number of remote children. $a_i$ is the transition taken $i$ steps back. All feature templates correspond to binary features.

$G_g$ are trees. Punctuation marks are excluded from the evaluation, but not from the datasets.

**Conversions.** We explore two conversion scenarios: one into (possibly discontinuous) constituency trees, and one into CoNLL-style dependencies. In the first setting we experiment with UPARSE, the only transition-based constituency parser, to our knowledge, able to parse trees with discontinuous constituents. In the second setting we use the MaltParser with arc-standard and arc-eager transition sets (Nivre et al., 2007),[7] and the stack LSTM-based arc-standard parser (Dyer et al., 2015). For MaltParser, we try both SVM and perceptron classifiers, reporting results obtained with SVM (about 1% F-score higher). Default settings are used in all cases. We do not use existing dependency DAG parsers since we could not obtain their code. We note that UPARSE uses beam search by default, with a beam size of 4, where the other parsers use greedy search.

Upper bounds for the conversion-based methods are computed by applying the conversion and inverse conversion on the gold standard graphs and comparing them to the original gold standard.

**BSP.** We train BSP for 16 iterations, using MINUPDATE = 5 and IMPORTANCE = 2, doubling weight updates for gold SWAP transitions to address the sparseness of discontinuous structures, as in Maier (2015). We train BSP both with and without remote edge transitions, to allow better comparison with conversion-based methods that only predict trees.

## 6 Results

Table 2 presents the results of our main experiments, as well as upper bounds for the conversion-based methods. BSP obtains comparable F-scores to MaltParser and UPARSE in terms of primary edges, but unlike them, is able to predict some of the remote edges too. Removing remote edge transitions from BSP does not change results considerably on the primary edges, improving them by 0.9% F-score in the in-domain setting, but reduces them by the same amount when applied out-of-domain. Out-of-domain results are largely comparable with the in-domain results, demonstrating robustness by BSP to domain variation.

The LSTM parser obtains the highest primary

---

[7]Preliminary experiments with non-projective variants of MaltParser yielded lower scores than projective ones, and were thus discarded from the final evaluation.

F-score, with a considerable margin. Importantly, it obtains 9.1% F-score higher than the arc-standard MaltParser, differing from it only in its classifier. This suggests that applying a similar approach to BSP is likely to improve results, and further underscores the effectiveness of transition-based methods for BSS parsing.

The conversion to constituency format only removes remote edges, and thus obtains a perfect primary edge score. The conversion to dependency format loses considerably more information, since all non-terminal nodes are lost and have to be reconstructed by a simple rule-based inverse conversion. Both conversions yield zero scores on remote edges, since these are invariably removed when converting to trees.

|  | Primary | | | Remote | | |
|---|---|---|---|---|---|---|
|  | LP | LR | LF | LP | LR | LF |
| *Constituency Tree Conversion* | | | | | | |
| UPARSE | 64 | 67.3 | 65.4 | – | 0 | 0 |
| Upper Bound | 100 | 100 | 100 | – | 0 | 0 |
|  | | | | | | |
| *Dependency Tree Conversion* | | | | | | |
| Malt$_{arc\text{-}standard}$ | 63.4 | 57.3 | 60.1 | – | 0 | 0 |
| Malt$_{arc\text{-}eager}$ | 63.9 | 57.9 | 60.5 | – | 0 | 0 |
| LSTM | **73.2** | **66.2** | **69.2** | – | 0 | 0 |
| Upper Bound | 93.8 | 83.7 | 88.4 | – | 0 | 0 |
|  | | | | | | |
| *Direct Approach* | | | | | | |
| BSP | 62.4 | 56 | 59 | 15.3 | 11.8 | 13.3 |
| BSP$_{Tree}$ | 63.8 | 56.5 | 59.9 | – | 0 | 0 |
|  | | | | | | |
| *Out-of-domain* | | | | | | |
| BSP | 60.6 | 53.9 | 57.1 | 20.2 | 10.3 | 13.6 |
| BSP$_{Tree}$ | 60.2 | 52.8 | 56.2 | – | 0 | 0 |

Table 2: Main experimental results in percents (on the *Wiki* test set, except for the bottom part). Columns correspond to labeled precision, recall and F-score for the different parsers, for both primary (left-hand side) and remote (right-hand side) edges. Top: results for UPARSE after conversion to constituency tree annotation. Upper middle: results for the Malt-Parser arc-eager and arc-standard, and the LSTM parser, after conversion to dependency tree annotation. Lower middle: results for our BSP, when trained on the complete UCCA DAGs (BSP), and when trained on UCCA trees, obtained by removing remote edges (BSP$_{Tree}$). Bottom: results for BSP and BSP$_{Tree}$ when tested on out-of-domain data (*20K Leagues*).

**Feature Ablation.** To evaluate the relative impact of the different feature sets on BSP, we remove a set of features at a time, and evaluate the resulting parser on the development set (Table 3). Almost all feature sets have a positive contribution to the primary edge F-score, or otherwise to the prediction of remote edges. **unigrams** and **bigrams** features are especially important, and the **ratio** feature greatly improves recall on primary edges. **disco** features have a positive contribution,

likely to be amplified in languages with a higher percentage of discontinuous units, e.g. German.

|  | Primary | | | Remote | | |
|---|---|---|---|---|---|---|
|  | LP | LR | LF | LP | LR | LF |
| BSP | 62.6 | 55.7 | 58.9 | 20 | 12.9 | 15.7 |
| BSP−**unigrams** | 62.5 | 52.6 | 57.1 | 18.9 | 10.1 | 13.2 |
| BSP−**bigrams** | 59.8 | 50.0 | 54.4 | 18.2 | 12.2 | 14.6 |
| BSP−**trigrams** | 63.7 | 55.0 | 59.0 | 20.7 | 12.0 | 15.2 |
| BSP−**separator** | 62.9 | 53.5 | 57.8 | 17.8 | 11.7 | 14.1 |
| BSP−**extended** | 62.9 | 52.8 | 57.4 | 17.4 | 11.5 | 13.9 |
| BSP−**disco** | 63.6 | 53.6 | 58.2 | 19.7 | 11.5 | 14.5 |
| BSP−**counts** | 63.3 | 52.8 | 57.6 | 14.6 | 10.6 | 12.3 |
| BSP−**edges** | 63.5 | 54.9 | 58.9 | 23.6 | 14.5 | 17.9 |
| BSP−**history** | 63.1 | 53.2 | 57.8 | 23.7 | 14.5 | 18.0 |
| BSP−**remote** | 63.5 | 53.2 | 57.9 | 17.2 | 10.6 | 13.1 |
| BSP−**ratio** | 63.7 | 48.3 | 55.0 | 25.4 | 13.6 | 17.7 |

Table 3: Results on the development set for the feature ablation experiment, in percents. The first row corresponds to BSP as in the main experiment. In each of the other rows, one feature set is excluded. Columns are the same as in Table 2. See Section 4 and Figure 3 for the feature set definitions.

# 7 Conclusion

We have introduced the first parser that supports multiple parents, non-terminal nodes and discontinuous units. We further explored a conversion-based parsing approach to assess the ability of existing technology to address the task. The work makes a further contribution by first experimenting on UCCA parsing. Results show that UCCA can be parsed with 69.2% primary F-score, and suggest means for improvement by taking an LSTM-based approach to the local classifier of BSP. The quality of the results is underscored by UCCA's inter-annotator agreement (often taken as an upper bound) of 80–85% F-score on primary edges (Abend and Rappoport, 2013).

While much recent work focused on semantic parsing of different types, the relations between the different representations have not been clarified. We intend to further explore conversion-based parsing approaches, including different target representations and more sophisticated conversion procedures (Kong et al., 2015), to shed light on the commonalities and differences between representations, suggesting ways to design better semantic representations. We believe that UCCA's merits in providing a cross-linguistically applicable, broad-coverage annotation will support ongoing efforts to incorporate deeper semantic structures into a variety of applications, such as machine translation (Jones et al., 2012) and summarization (Liu et al., 2015).

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
