# Peer review of "Broad-Coverage Semantic Parsing: A Transition-Based Approach"

_CoNLL 2016 — decision unknown_

[Official Review · Reviewer 1 · rating 3 · confidence 5]
soundness 4 · originality 3 · clarity 4 · impact 3 · substance 4 · appropriateness 5 · meaningful comparison 4 · replicability 5 · presentation format Oral Presentation

This paper presents a transition-based graph parser able to cope with the rich
representations of a semantico-cognitive annotation scheme, instantiated in the
UCCA corpora. The authors start first by exposing what, according to them,
should cover a semantic-based annotation scheme: (i) being graph-based
(possibility for a token/node of having multiple governors) (2) having
non-terminal nodes (representing complex structures â syntactic -: coordinate
phrases, lexical: multiword expression) and (3) allowing discontinuous elements
(eg. Verbs+particules). Interestingly, none of these principles is tied to a
semantic framework, they could also work for syntax or other representation
layers. The authors quickly position their work by first introducing the larger
context of broad-coverage semantic parsing then their annotation scheme of
choice (UCCA).              They then present 3 sets of parsing experiments: (i) one
devoted to phrase-based parsing using the Stanford parser and an UCCA to
constituency conversion, (ii) one devoted to dependency parsing using an UCCA
to dependency conversion and finally (iii) the core of their proposal, a  set
of experiments showing that their transition-based graph parser is suitable for
direct parsing of UCCA graphs.

I found this work interesting but before considering a publication, I have
several concerns with regards to the methodology and the empirical
justifications:

The authors claimed that there are the first to propose a parser for a
semantically-oriented scheme such as theirs. Of course, they are. But with all
due respect to the work behind this scheme, it is made of graphs with a various
level of under-specified structural arguments and semantically oriented label
(Process, state) and nothing in their transition sets treats the specificities
of such a graph. Even the transitions related to the remote edges could have
been handled by the other ones assuming a difference in the label set itself
(like adding an affix for example). If we restrict the problem to graph
parsing, many works post the 2014-2015 semeval shared tasks (Almeda and
Martins, 2014,2015 ; Ribeyre et al, 2014-2015) proposed an extension to
transition-based graph parser or an adaptation of a higher-model one, and
nothing precludes their use on this data set.  Itâs mostly the use of a
specific feature template that anchors this model to this scheme (even though
itâs less influencial than the count features and the unigram one). Anyway,
because the above-mentioned graph-parsers are available [1,2] I donât
understand why they couldnât be used as a baseline or source of comparisons.
Regarding the phrase-based  experiments using uparse, it could have been also
validated by another parser from Fernandez-Gonzales and Martins (2015) which
can produce LCFRS-like parsing as good as Uparse (ref missing when you first
introduced uparse).  

Because this scheme supports a more abstract view of syntaxico-semantic
structures than most of the SDP treebanks, it would have been important to use
the same metrics as in the related shared task. At this point in the field,
many systems, models and data set are competing and I think that the lack of
comparison points with other models and parsers is detrimental to this work as
whole. Yet I found it interesting and because weâre at crossing time in term
of where to go next, I think that this paper should be discussed at a
conference such as ConLL.

Note in random order
-         please introduce the âgrounded semanticâ before page 2, you use
that phrase before
-         why havenât you try to stick to constituent-tree with rich node
labels and propagater traces and then train/parse with the Berkeley parser? It
could have been a good baseline. 
-         The conversion to surface dependency trees is in my mind useless: you
loose too many information, here a  richer conversion such as the one from
âSchluter et al, 2014, Semeval SDP) should have been used.
-         Can you expand on âUCCA graphs may contains implicit unit that have
no correspondent in the textâ  or provide a ref or an example.
-         You mentioned other representations such as MRS and DRT, this raises
the fact that your scheme doesnât seem to allow for a modelling of quantifier
scope information. Itâs thus fully comparable to other more syntax-oriented
scheme. Itâs indeed more abstract than DM for example and probably more
underspecified than the semantic level of the PCEDT but how much? How really
informative is this scheme and how really âparsableâ is it? According to
your scores, it seems âharderâ but an  error analysis would have been
useful.
- As I said before, the 3 principles you devised could apply to a lot of
things,  they look a bit ad-hoc to me and would probably need to take place in
a much wider (and a bit clearer) introduction. What are you trying to argue
for: a parser that can parse UCCA? a model suitable for semantic analysis ? or
a semantic oriented scheme that can actually be parsable?  you're trying to say
all of those in a very dense way and it's borderline to be be confusing.

[1] http://www.corentinribeyre.fr/projects/view/DAGParser
[2] https://github.com/andre-martins/TurboParser and
https://github.com/andre-martins/TurboParser/tree/master/semeval2014_data

[Official Review · Reviewer 2 · rating 2 · confidence 4]
soundness 3 · originality 3 · clarity 3 · impact 3 · substance 2 · appropriateness 5 · meaningful comparison 2 · replicability 4 · presentation format Oral Presentation

The paper presents the first broad-coverage semantic parsers for UCCA, one
specific approach to graph-based semantic representations. Unlike CoNLL
semantic dependency graphs, UCCA graphs can contain "nonterminal" nodes which
do not represent words in the string. Unlike AMRs, UCCA graphs are "grounded",
which the authors take to mean that the text tokens appear as nodes in the
semantic representation. The authors present a number of parsing methods,
including a transition-based parser that directly constructs UCCA parses, and
evaluate them.

Given that UCCA and UCCA-annotated data exist, it seems reasonable to develop a
semantic parser for UCCA. However, the introduction and background section hit
a wrong note to my ear, in that they seem to argue that UCCA is the _only_
graph-based semantic representation (SR) formalism that makes sense to be
studied. This doesn't work for me, and also seems unnecessary -- a good UCCA
parser could be a nice contribution by itself.

I do not entirely agree with the three criteria for semantic representation
formalisms the authors lay out in the introduction. For instance, it is not
clear to me that "nonterminal nodes" contribute any expressive capacity. Sure,
it can be inconvenient to have to decide which word is the head of a
coordinated structure, but exactly what information is it that could only be
represented with a nonterminal and not e.g. with more informative edge labels?
Also, the question of discontinuity does not even arise in SRs that are not
"grounded". The advantages of "grounded" representations over AMR-style ones
did not become clear to me. I also think that the word "grounded" has been used
for enough different concepts in semantics in the past ten years, and would
encourage the authors to find a different one ("anchored"? "lexicalized"?).
Thus I feel that the entire introductory part of the paper should be phrased
and argued much more carefully.

The parser itself seems fine, although I did not check the details. However, I
did not find the evaluation results very impressive. On the "primary" edges,
even a straightforward MaltParser outperforms the BSP parser presented here,
and the f-scores on the "remote" edges (which a dependency-tree parser like
Malt cannot compute directly) are not very high either. Furthermore, the
conversion of dependency graphs to dependency trees has been studied quite a
bit under the name "tree approximations" in the context of the CoNLL 2014 and
2015 shared tasks on semantic dependency parsing (albeit without "nonterminal"
nodes). Several authors have proposed methods for reconstructing the edges that
were deleted in the graph-to-tree conversion; for instance, Agic et al. (2015),
"Semantic dependency graph parsing using tree approximations" discuss the
issues involved in this reconstruction in detail. By incorporating such
methods, it is likely that the f-score of the MaltParser (and the LSTM-based
MaltParser!) could be improved further, and the strength of the BSP parser
becomes even less clear to me.